# Study of the OLR Anomalies before the 2023 Turkey M7.8 Earthquake

Jun Liu [1], Jing Cui [2,*], Ying Zhang [3], Jie Zhu [4], Yalan Huang [2], Lin Wang [1] and Xuhui Shen [5]

[1] School of Ecological Environment, Institute of Disaster Prevention, Sanhe 065201, China; liujun.3s@cidp.edu.cn (J.L.); 21661332@sr.cidp.edu.cn (L.W.)
[2] National Institute of Natural Hazards, Ministry of Emergency Management of the People's Republic of China, Beijing 100085, China; huangyalan22@mails.ucas.ac.cn
[3] Department of Geophysics, School Earth and Space Sciences, Peking University, Beijing 100091, China; zhangying2016@radi.ac.cn
[4] China Earthquake Networks Center, Beijing 100045, China; zhujie@seis.ac.cn
[5] National Space Science Center, Chinese Academy of Sciences, Beijing 100190, China; xuhuishen@ninhm.ac.cn
* Correspondence: jingcui@ninhm.ac.cn; Tel.: +86-01062846770

**Abstract:** Using the model of the additive tectonic stress from celestial tide-generating force, we studied the relationship between the seismogenic structure and celestial tide-generating stress in the M7.8 Turkey earthquake on 6 February 2023. We analyzed the daily continuous variation characteristics of OLR before and after the Turkey earthquake and discussed the correlation characteristics of tidal stress, OLR, and the earthquake. The results showed that the observed OLR anomaly according to the tidal stress variation cycle "C" (1–15 February) presented a phase change in time, which was synchronized with a continuous trough-to-peak change in the additional tectonic main pressure stress. The spatial distribution of OLR anomalies was mainly concentrated in the southwest section of the East Anatolian Fault Zone, which indicates that seismic tectonic movements were the main causes of OLR anomaly variation during this earthquake. An OLR anomaly change was related to this M7.8 "Swarm Type" of earthquake in Turkey. Impending earthquake OLR anomalies represent that the stress of the seismogenic structure in the seismogenic region has entered a critical state, which can provide stress monitoring and a seismogenic region indication for earthquakes induced by tidal force. The change cycle of the celestial tide-generating force provides a time indication for the identification of seismic thermal anomalies, and it indicates that the combination of the additional tectonic stress of the tidal force and the change of OLR anomaly has value for the research on the short-impending earthquake precursor.

**Keywords:** earthquake; OLR; short-impending anomaly; tidal stress; Turkey

## 1. Introduction

Earthquakes are the result of the tectonic deformation and rupture of faults under tectonic stress, but in situ stress cannot be directly observed at present [1]. By observing the change in heat-related physical quantities, tectonic stress information can be obtained indirectly [2,3]. The occurrence of atmospheric thermal anomalies before earthquakes has been widely recognized by scholars [4,5]. Scholars from the former Soviet Union [3] found that the phenomenon of short-impending infrared radiation enhancement occurs at the intersection of faults before earthquakes, which introduced a new way to study earthquakes using satellite thermal infrared remote sensing technology. In the last 30 years, many scholars have explored the application of satellite remote sensing technology in seismic activity monitoring and earthquake precursor research, and many achievements have been made in the studies on the generation mechanism of thermal anomalies [3,6–13] and the extraction algorithm of seismic thermal anomalies [14–21]. Pre-earthquake thermal anomalies have also been verified in many case studies [22,23].

In order to scientifically elucidate the physical mechanism of infrared thermal anomalies in tectonic earthquakes, both domestic and foreign scholars have conducted extensive exploration and research using various methods such as air field experiments and rock-loading experiments. Consequently, several hypotheses including the gas-thermal [3,6], stress-induced heat [7–9], "P-hole" [11], and latent heat release due to radon decay theories [13] have been proposed, but remain inconclusive. However, considering the sources of heat during the earthquake preparation process, it is evident that with the progressive accumulation of tectonic stress, mechanical energy, along with other forms of physical and chemical energy, are generated deep within the Earth's interior. Subsequently, a significant portion of this energy is converted into heat, which is then released through various mechanisms in the form of infrared electromagnetic radiation [14]. At present, the methods for pre-earthquake thermal anomaly extraction using satellite infrared information include the background field difference analysis method [15], the robust satellite data analysis technique (RST) [16,17,24,25], the eddy field calculation mean algorithm [26,27], the wavelet-power spectrum method [21], the spatio-temporally weighted two-step method [28], etc. These methods require many years of historical data. Moreover, the normal background field established by the multi-year data-averaging algorithm based on statistical principles will mask small fluctuations in the data and miss the weak thermal anomaly signal before the earthquake. At the same time, due to the uncertainty of the selection time (5 years, 10 years, etc.), the different background fields will present different results [29–31].

The occurrence of earthquakes is a manifestation of internal tectonic movements within the Earth. However, as the Earth is not an isolated celestial body, its movement is inevitably influenced by macroscopic celestial movements. One external mechanical factor that affects the Earth's movement is sun–moon tidal forces. This force generates solid tides in the Earth's interior, leading to periodic tidal stresses with magnitudes in the order of $10^3$ Pa. Although these tidal stresses are much smaller than seismic stress drops, their cyclic loading rate exceeds that of tectonic stress accumulation by two orders of magnitude [32,33]. The stress in the Earth's crust results from both tectonic and tidal forces acting upon it. When seismic tectonic stress reaches a critical state, causing rock sliding, and if a rapidly changing tidal stress is superimposed in an appropriate direction, it may trigger an earthquake [32,34–37]. In order to further study the dynamic change of tidal stress in the process of time series and its relationship with the seismic process, Ma et al. [38] proposed a model of the additive tectonic stress from celestial tide-generating force (abbreviated as ATSCTF). This model calculates the components of tidal stress ($\partial P$, $\partial T$, and $\partial N$) generated by the celestial tide-generating force at the epicenter along the direction of the main pressure stress P-axis, the tension stress T-axis, and the vertical stress N-axis. The induced effects of additional tectonic stresses on seismic faults are classified into three types [39,40]. By drawing time-series variation curves for each component according to the ATSCTF models, this provides a clear temporal context with mechanical significance for studying the thermal anomalies associated with earthquakes and enables the monitoring of tectonic stress states through the observation of thermal anomalies caused by tide-induced earthquakes [41,42].

At present, when infrared remote sensing data are used to study seismic and tectonic activities, the cloud interference often hinders the observation of temperature anomalies before earthquakes. Outgoing long-wave radiation (OLR) represents the energy density of electromagnetic waves emitted into outer space via Earth's atmospheric system, and the radiation physical quantity most directly reflects the underlying surface properties and energy variation parameters. Moreover, its band (10.5–12.5 μm) is concentrated in the atmospheric window and minimally interferes with clouds [31]. Therefore, OLR data were selected for this research. Aiming to study the 2023 "earthquake swarm type" M7.8 earthquake in Turkey, we combined the celestial tide-generating force and OLR data to study the correlation between two physical parameters and earthquakes and discussed the process of earthquake breeding and occurrence.

## 2. Study Area

According to the U.S. National Earthquake Information Center (NEIC), at 01:17 (UTC) on 6 February 2023 (4:18 local time on 6 February), a 7.8 magnitude earthquake with a focal depth of 17.9 km occurred 26 km east of Nurdaği in Gaziantep province, south-central Turkey, near the northern Syrian border. The epicenter was located at 37.17°N and 37.03°E, and aftershocks continued. Nine hours later, at 10:24 (UTC) (13:25 local time), a 7.5 magnitude earthquake occurred again about 96 km northeast of the epicenter of the 7.8 magnitude earthquake. The epicenter was located at 38.02°N, 37.20°E. The focal depth of this earthquake was 10 km. These two powerful earthquakes caused many casualties and property losses in Turkey, and this is the worst disaster to hit the country in more than 80 years, arousing the attention of the international community.

Most of Turkey is located on the Anatolian block where the Eurasian plate, the Arabian plate, and the African plate meet, and its geological structure is very complex. Two huge strike-slip fault zones, the East Anatolian Fault and the North Anatolian Fault [36], developed in Turkey under the influence of the tectonic compression of multiple plates. The two fault zones converge in eastern Turkey (Figure 1). Both earthquakes occurred near the northeasterly extending East Anatolian Fault zone, which is one of the major intracontinental conversion faults in the Eastern Mediterranean region. Together with the North Anatolian right-rotation Fault zone, it accommodated the westward compression tectonic movement of the Anatolian block. This fault zone is also controlled by the northwest subduction of the Arabian plate. The rupture of this earthquake was more than 400 km long.

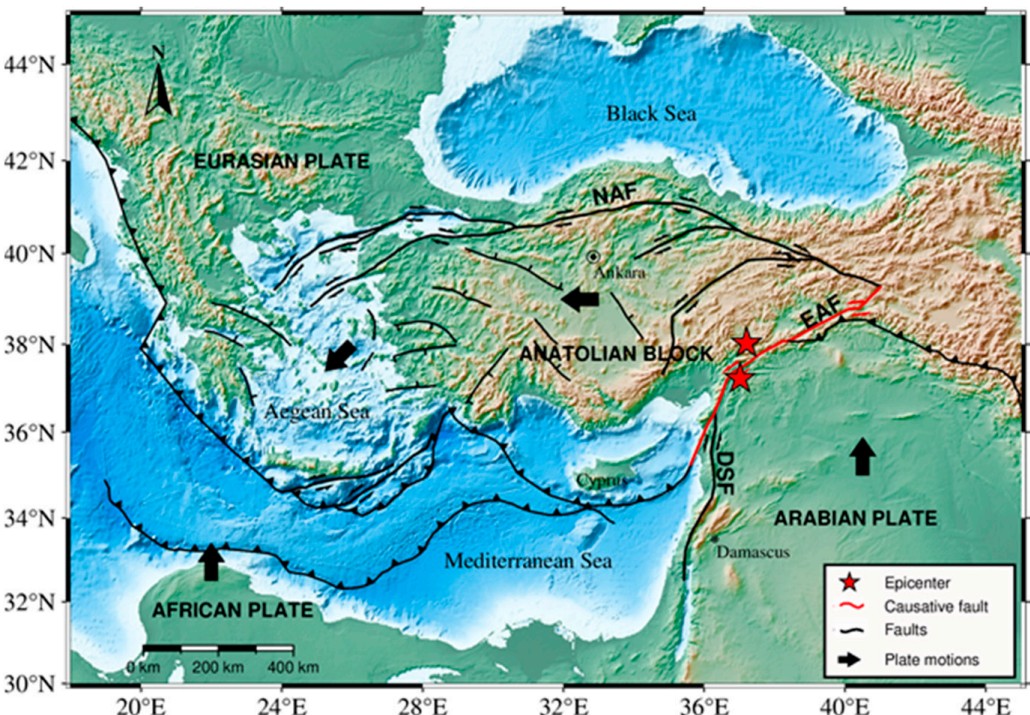

**Figure 1.** Distribution of major active faults in Turkey earthquake epicenter and study area [43,44]. NAF: North Anatolian Fault; EAF: East Anatolian Fault; DSF: Dead Sea Fault.

## 3. Materials and Methods

### 3.1. Calculation of the Additive Tectonic Stress from the Celestial Tide-Generating Force

To analyze the influence of additive tectonic stress from the celestial tide-generating force on the seismogenesis fault, the tidal stress component in the axis coordinate system of seismic stress is calculated. It is generated by the celestial tide-generating force of the sun and the moon at the epicenter along the directions of the main pressure stress

P-axis, the tension stress T-axis, and the vertical stress N-axis. This is performed to explore the relationship between the size, direction, and time of seismic occurrence and the region of seismic occurrence [39] and provide timely guidance for the selection of the OLR abnormal background.

The calculation of tidal stress components at the epicenter is divided into three steps. Firstly, the tidal stress components separately generated by the moon and the sun at the source in the spherical coordinate system are calculated according to Kelvin's method [45]. Then, the tidal stress components from the spherical coordinate system are converted to the rectangular coordinate system. Finally, the tidal stress component in the rectangular coordinate system of the earthquake source generated by the sun and the moon is converted into the stress component in the main axis coordinate system of the earthquake stress, and then the tidal stress components at the source along the P-axis, T-axis, and N-axis, $\partial_P$, $\partial_T$, and $\partial_N$, respectively, are obtained [40]. The converted formula is as follows:

$$\begin{cases} \partial_P = \sigma_{xx} B_{PX}^2 + \sigma_{yy} B_{PY}^2 + \sigma_{zz} B_{PZ}^2 + 2\sigma_{xy} B_{PX} B_{PY} + 2\sigma_{yz} B_{PY} B_{PZ} + 2\sigma_{zx} B_{PZ} B_{PX} \\ \partial_T = \sigma_{xx} B_{TX}^2 + \sigma_{yy} B_{TY}^2 + \sigma_{zz} B_{TZ}^2 + 2\sigma_{xy} B_{TX} B_{TY} + 2\sigma_{yz} B_{TY} B_{TZ} + 2\sigma_{zx} B_{TZ} B_{TX} \\ \partial_N = \sigma_{xx} B_{NX}^2 + \sigma_{yy} B_{NY}^2 + \sigma_{zz} B_{NZ}^2 + 2\sigma_{xy} B_{NX} B_{NY} + 2\sigma_{yz} B_{NY} B_{NZ} + 2\sigma_{zx} B_{NZ} B_{NX} \end{cases} \tag{1}$$

In Equation (1), $\sigma_{ij}(i, j = x, y, z)$ represents the 9 tidal stress components generated by the joint action of the moon and the sun in the source coordinate system; $B_{i,j}(i = P, T, N, j = X, Y, Z)$ is the coordinate transformation matrix consisting of the rotation angles $H$, $Q$, and $V$ in the transformation between the source rectangular coordinate system and the seismic stress main axis coordinate system, which are expressed as:

$$\begin{cases} B_{PX} = -\sin H \cos Q \sin V - \sin Q \cos V \\ B_{PY} = -\sin H \sin Q \sin V + \cos Q \cos V \\ B_{PZ} = \sin V \cos H \\ B_{TX} = -\sin H \cos Q \cos V + \sin Q \sin V \\ B_{TY} = -\sin H \sin Q \cos V - \cos Q \sin V \\ B_{TY} = \cos H \cos V \\ B_{NX} = \cos H \cos Q \\ B_{NY} = \cos H \sin Q \\ B_{NZ} = \sin H \end{cases} \tag{2}$$

In Equation (2), $H$, $Q$, and $V$ are the rotation angles of the coordinate transformation, which can be obtained from the source parameters (inclination and strike of the P-axis and T-axis).

### 3.2. OLR Data Processing

OLR is the energy density of electromagnetic waves emitted into outer space via Earth's atmospheric system, also known as the thermal radiation flux density, which is measured in $W/m^2$. It is measured using an NOAA polar orbit satellite load radiation measuring instrument, which scans the Earth and the atmosphere in the infrared window channel (10.5–12.5 μm) and detects long-wave radiation emitted from the ground.

This research adopted OLR data taken from http://www.emc.ncep.noaa.gov (accessed on 5 May 2023), which have a temporal resolution of 24 h and a spatial resolution of $1 \times 1°$, covering a total of $360 \times 181$ grids worldwide. Since OLR is based on remote sensing output results in the infrared band, its band approximates the long-wave atmospheric window, for which atmospheric decay is weak, and its band is close to the surface long-wave radiation; additionally, it is sensitive to temperature changes in the sea surface and near the ground. Therefore, it is suitable for monitoring some geo-hazard signs associated with "thermal" genesis phenomena.

The OLR values of the grid points (inter-daily scale) in the study area are calculated using Formula (3) to extract the changing characteristics of the OLR data before and every day after the earthquake. Subsequently, the numerical distribution of each grid point representing information in the radiation enhancement area is obtained.

$$\Delta T_i(lon, lat) = T_i(lon, lat) - T_{background}(lon, lat) \tag{3}$$

$\Delta T_i(lon, lat)$ represents the incremental OLR value in a Gaussian grid $(lon, lat)$; $T_i(lon, lat)$ represents the OLR value in a Gaussian grid $(lon, lat)$; $T_{background}(lon, lat)$ represents the OLR value in a Gaussian grid $(lon, lat)$ on a fixed background day; $lon$ is the geodetic longitude, $lon = 1, 2, \ldots, 360$; $lat$ is the geodetic latitude, $lat = 1, 2, \ldots, 180$; $i$ is the date. The background day is determined based on the inflection point of the period for additional tectonic stress caused by tidal-generating forces.

## 4. Results

### 4.1. Analysis of the Additive Tectonic Stress from the Celestial Tide-Generating Force Change in Turkey Earthquake

The tidal stress generated by the sun and the moon exhibits periodic and continuous changes, with its magnitude being influenced by the focal mechanism of earthquakes. The turning point in the period of tidal stress change represents a position where the impact of tidal forces on the tectonic environment alters, and this period is typically divided based on such turning points (wave crests or troughs) [41]. According to the focal mechanism solution provided by USGS (https://earthquake.usgs.gov/) (accessed on 19 February 2023), we calculated the additional tectonic stress component ($\partial P$, $\partial T$, and $\partial N$) at the epicenter of the Turkey earthquake for the period from 1 January to 1 March 2023 using the ATSCTF model [38]. The results were then plotted as a time-dependent curve in Figure 2. The figure illustrates that the additional stress components $\partial T$ and $\partial N$ generated by tidal forces from celestial bodies were relatively small along the tension stress T-axis and the vertical stress N-axis, whereas the component $\partial P$ generated in the direction of the main pressure stress P-axis was significant. Consequently, the period of tidal stress variation can be divided into four consecutive periods (marked as A, B, C, and D) based on the inflection point (trough) of the change curve for the $\partial P$ value. Period A corresponds to 3–17 January, period B corresponds to 18–31 January, period C corresponds to 1–15 February, and period D corresponds to 16 February–1 March. The initial day of each cycle serves as the foundational background day. The earthquake occurred in close proximity to the relative high point of the principal compressive stress $\partial P$ during its transition from trough to peak [40], suggesting that the celestial tide-generating force continuously amplified the positive pressure on the fault surface. When the active tectonic stress reaches a critical threshold, the tide-generating force may trigger destabilization and the rupture of seismic structures under stress, leading to earthquakes.

The M7.8 earthquake in Turkey occurred in cycle "C", but no earthquake occurred (http://www.ceic.ac.cn) (accessed on 20 June 2023) near the epicenter during similar phases of cycles A, B, and D, indicating that the celestial tide-generating force is only one of the important external factors that trigger and induce earthquakes. Only when the active tectonic stress reaches the critical point of rupture can the earthquake be induced [41]. To judge the stress accumulation in the seismogenic region and the size of the change in tectonic stress in the corresponding period, it is necessary to further analyze the spatial and temporal evolution characteristics of OLR against the background of the tidal force change period.

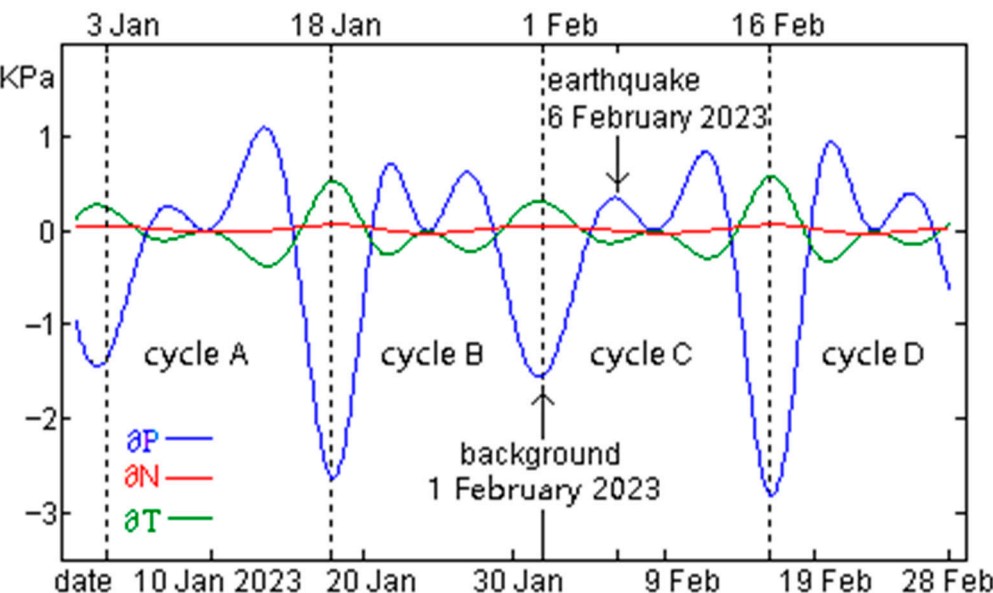

**Figure 2.** The additive tectonic stress components ∂P, ∂T, and ∂N changed over time before and after the earthquake in Turkey.

### 4.2. Spatial and Temporal Evolution Characteristics of OLR Anomaly in Turkey "Swarm Type" Earthquake

The choice of background date is critical when extracting OLR anomalies using Equation (3). Therefore, to avoid uncertainties in the calculation results due to differences in the duration of the background time in the statistical processing method of multi-year data [26,27], this paper provides a temporal guide with a clear physical meaning for the selection of the background date based on the cycle of the additive tectonic stress from the celestial tide-generating force variation. Taking the phase low point of the main stress value ∂P, with the variation curve as the reference (Figure 2), 3 January, 18 January, 1 February, and 16 February were selected as the background dates for the OLR data, respectively. According to Equation 3, the nighttime surface long-wave radiation values in the spatial range (20–50°N, 20–50°E) from 3 January to 1 March were subtracted from the OLR background values to obtain continuous day-by-day images of the OLR changes before and after the earthquakes in Turkey.

As shown in Figure 3c, within the study area, the epicenter of the Turkey earthquake and its adjacent areas showed a significantly anomalous OLR change before and after the earthquake after the low peak phase of the additive tectonic main stress from the celestial tide-generating force ∂P. Obvious OLR anomalies appeared south and southwest of the epicenter on 2 February, spanning the intersection of the Anatolian, African, and Arabian plates. The OLR anomalies were mainly distributed along both sides of the southwest end of the NE direction East Anatolia Fault Zone, indicating that the fault had a controlling effect on the OLR anomaly increase. On 3–4 February, the OLR anomaly declined, and its extent narrowed, but it was still symmetrically distributed on both sides of the NE direction East Anatolia Fault Zone. On 5 February, the OLR anomaly declined significantly in intensity and was spatially scattered in a northeast direction along the East Anatolian Fault Zone. The earthquake occurred on 6 February, when the additive tectonic stress from the celestial tide-generating main pressure stress ∂P reached its sub-peak phase, the positive pressure on the fault surface increased, and this promoted the fault stress state to reach a critical condition for rupture sliding and seismogenesis. The OLR anomaly was enhanced in the images again at the southwest end of the East Anatolian Fault Zone. After the mainshock, aftershock activity was frequent between 7 February and 15 February, and densely distributed at the southwest end of the East Anatolian Fault Zone. The frequent vibration of the earthquake formed numerous collapsed areas or large-scale surface ruptures, which became thermal

channels and released large amounts of heat, manifesting as the continuous expansion of the area and the magnitude of the OLR anomaly.

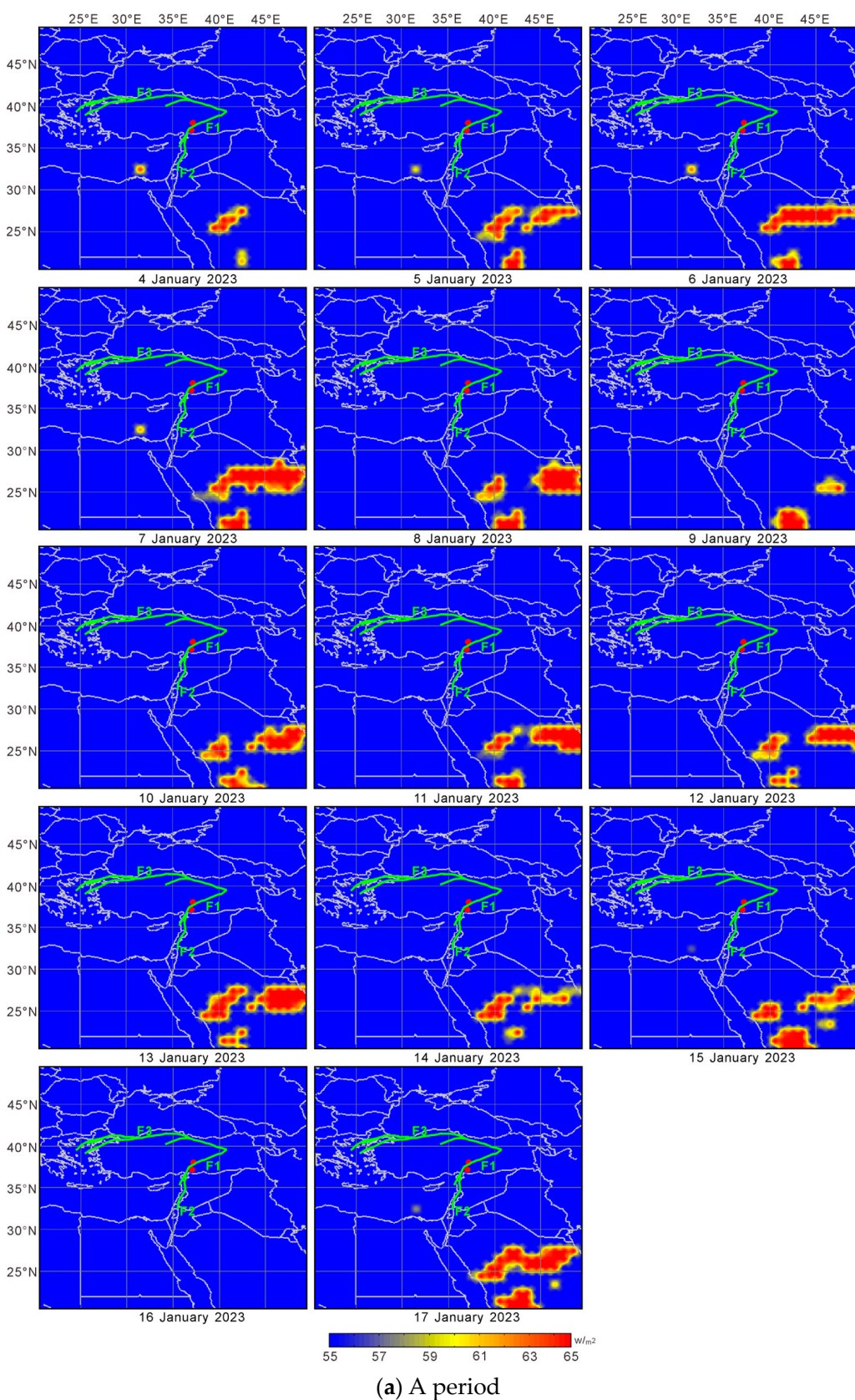

(**a**) A period

**Figure 3.** *Cont.*

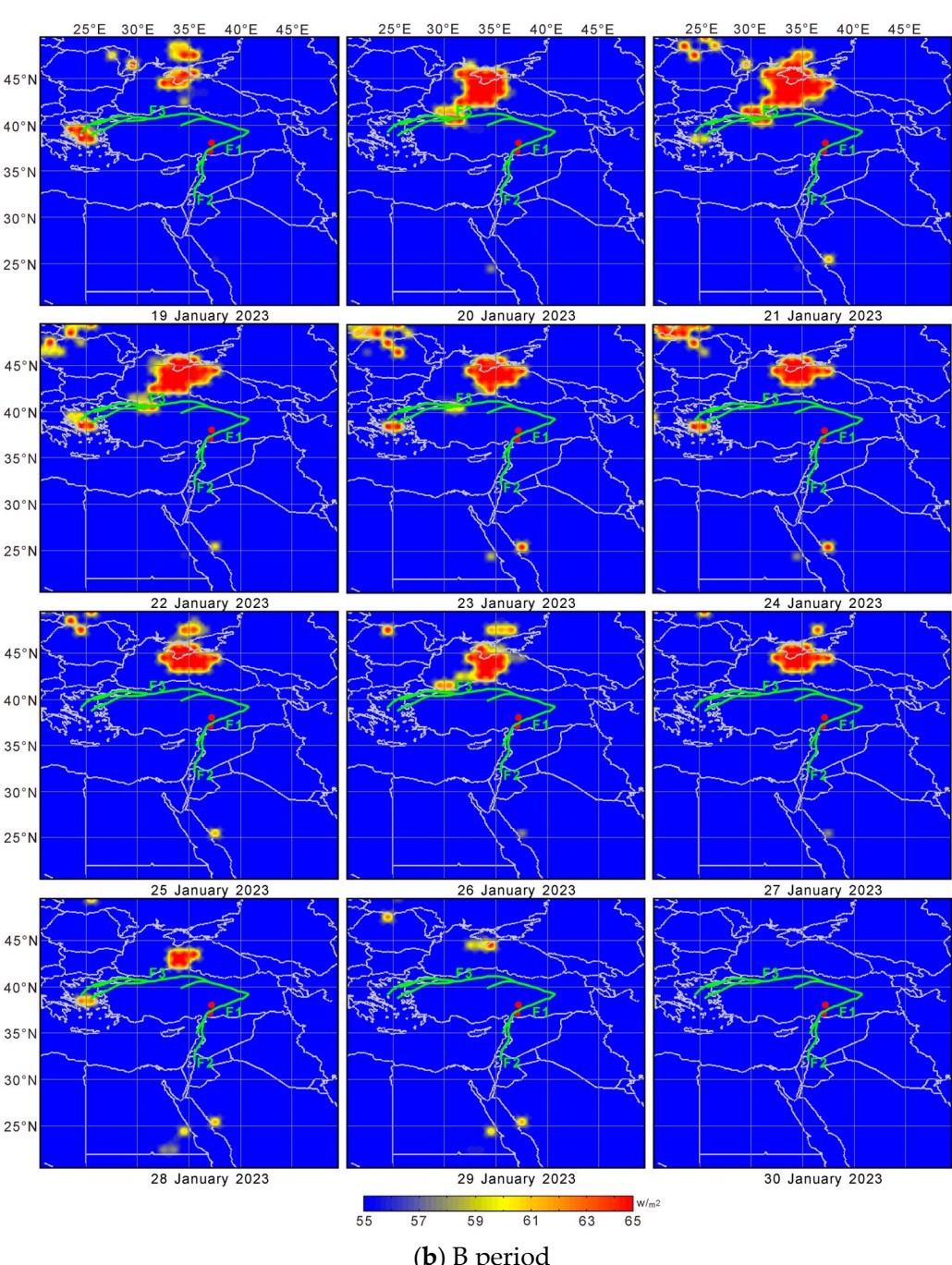

(**b**) B period

**Figure 3.** *Cont.*

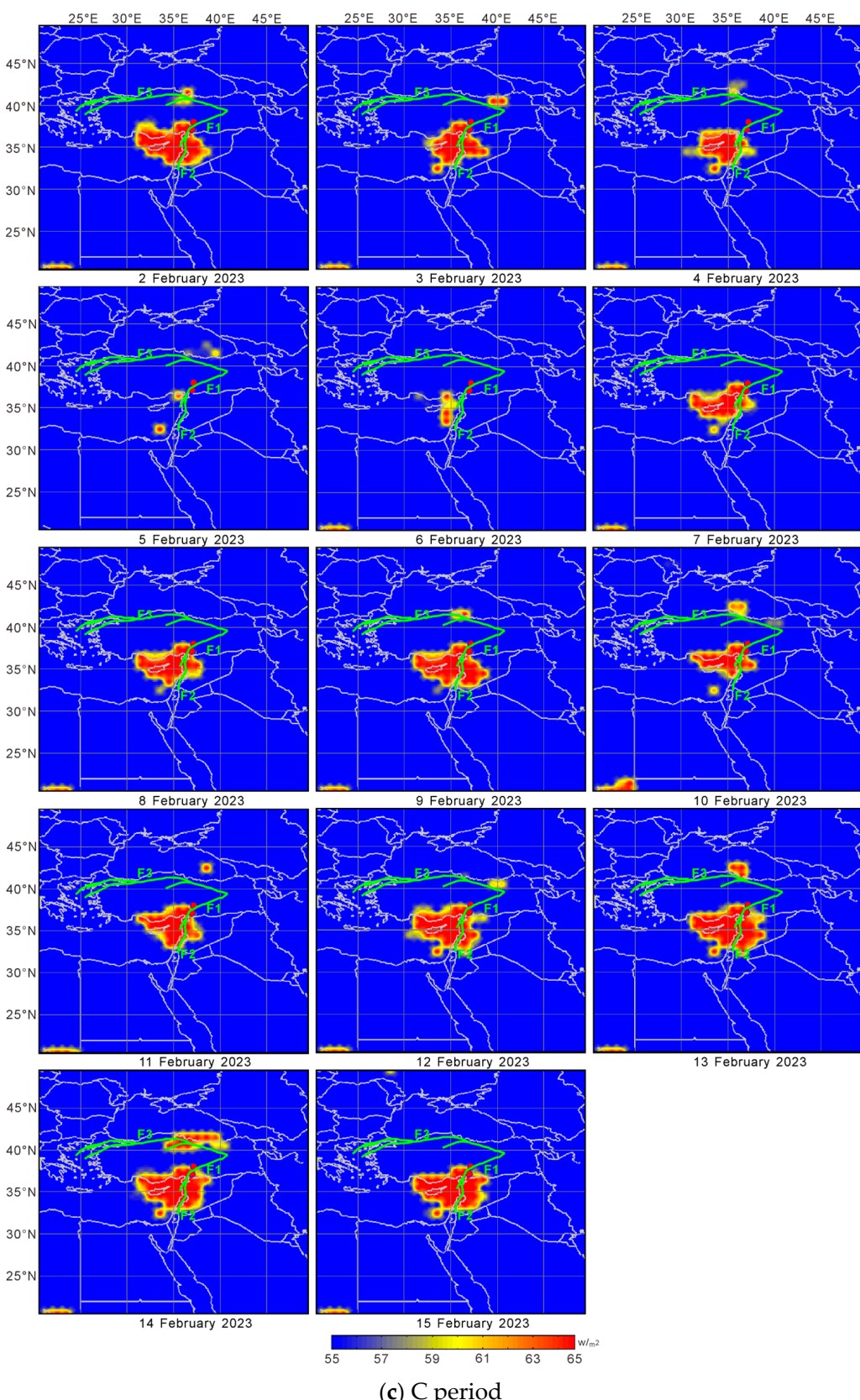

(**c**) C period

**Figure 3.** *Cont.*

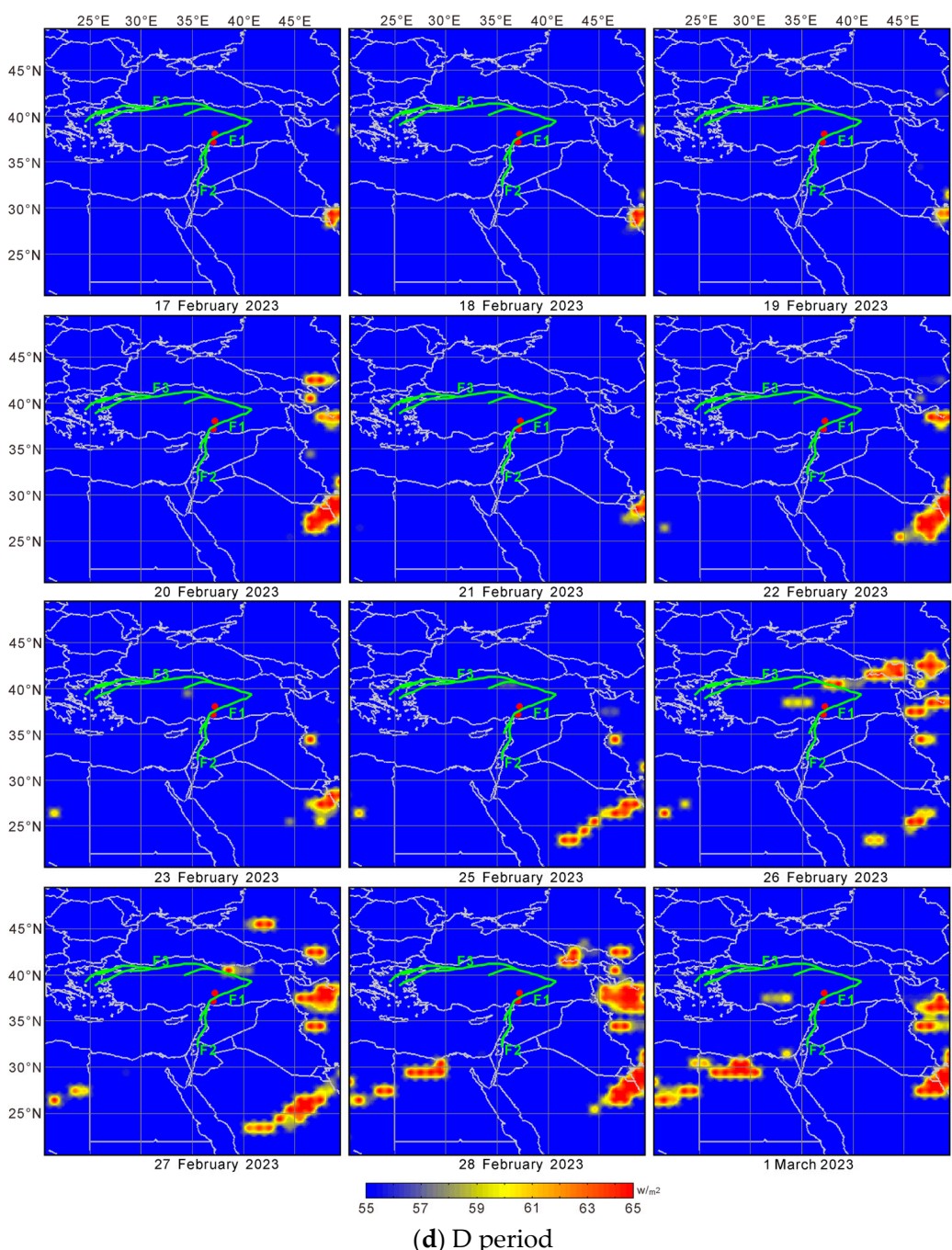

(**d**) D period

**Figure 3.** Images of the spatial and temporal evolution of ground-based long-wave radiation anomalies from the 7.8 magnitude earthquake in Turkey. The red dots are the epicenters. F1: East Anatolian Fault; F2: North Anatolian Fault; F3: Dead Sea Fault.

As shown in Figure 3a,b,d, there was no significant enhancement of OLR in and around the epicenter of the Turkey earthquake during the pre-earthquake tidal change cycle "A", "B", and the post-earthquake tidal change cycle "D", and no earthquake occurred. It should be noted that the OLR image for 24 February is missing from cycle "D" (Figure 3d) due to missing OLR data for that day.

## 5. Discussion

During the seismogenic cycle (cycle "C"), the earthquakes in Turkey occurred when the trough-to-peak main pressure stress change occurred, which was when the magnitude and

direction of the tidal stress suddenly changed, prompting the destabilizing rupture of the southwest section of the East Anatolian Fault in a high-stress state. However, no earthquake occurred near the epicenter in similar phases of other cycles of tidal stress change, indicating that tidal stress cannot directly trigger earthquakes and that the relationship between tidal stress and earthquakes cannot be studied in isolation from the regional tectonic setting.

From the analysis of Figure 3c, it can be seen that the OLR short-impending anomalies before and after the M7.8 earthquake in Turkey, which were observed synchronously according to the cycle changes in additive tectonic stress from the celestial tide-generating force, showed a change over time, undergoing the continuous evolution of initial warming (2 February)→anomalous decay (3–5 February)→mainshock (6 February)→aftershock→ continuous warming (7–15 February). This is consistent with the phase change in infrared radiation temperature in the rock stress rupture experiment [46]. The above process basically reflects the seismogenic process of micro-rupture→closure→major rupture (seismogenesis)→the rupture of active structures in all directions→tectonic adjustment (aftershock) during the seismogenesis. Combined with the theory of tidal force-induced earthquakes, when tectonic stresses have accumulated to enable rock rupture and sliding, earthquakes may be triggered by the superimposition of rapidly changing celestial tidal forces and additive tectonic stresses. It can be assumed that the OLR anomaly, which is synchronized with the cycle of tidal force change, is a thermal image representation of the seismic tectonic stress approaching the critical state of rupture, and that the tidal force induces earthquakes by changing the tectonic stress environment in the subsurface when the ground stress is sufficient. These "Swarm Type" earthquakes in Turkey are characterized by a high seismic frequency, a long duration of activity, and slow energy decay. After these two powerful earthquakes, the transition zone between the Dead Sea Fault Zone and the East Anatolian Fault Zone was characterized by high aftershock activity, with a total of 6000 aftershocks until 19 February, 40 of which were of magnitude 5–6, and 436 of which were of magnitude 4–5. Ongoing seismic activity induced the release of a substantial amount of thermal radiation through surface ruptures, which were visually depicted in the OLR change image as a persistent thermal radiation anomaly located in close proximity to the southwestern region of the East Fault belt. It has been shown that with the decay of the earthquake magnitude, the seismotectonics of the region will stabilize, and OLR anomalies will gradually disappear [41]. It has been observed in the rock-loading test that a transient cooling phenomenon occurred prior to rock instability, and this cooling anomaly served as a significant precursor to rock failure and instability, indicating the imminent macro failure of the rock [12,47,48]. Furthermore, the infrared seismic monitoring results revealed that apart from warming anomalies, certain earthquake events [49–51] exhibited a preceding cooling phenomenon before the occurrence of earthquakes. A similar process was also observed during the earthquakes in 2023 in Turkey. Therefore, we posit that the abnormal attenuation phenomenon of OLR prior to the earthquake on 5 February may signify a lock-in period before changes in rock stress lead to a rupture and could potentially serve as short-term imminent precursor information for earthquakes. The thermal anomalies associated with the Turkish earthquake were primarily distributed along the southwest section of the East Anatolian Fault Zone and northern section of the Dead Sea Fault, which aligned with the spatial distribution patterns of asperities identified in the East Anatolian Fault Zone [52]. These findings indicate that the OLR anomaly was influenced by the spatial distribution of seismogenic faults and tectonic activities, while representing thermal image manifestations of tectonic stress changes during earthquake preparation. The spatial and temporal evolutions of OLR shown in Figure 3c were consistent with the characteristics of seismic thermal anomaly expression, and it is suggested that there was a correlation between this OLR anomaly change and seismic activity.

To help verify the method's recognition rate, we achieved a coverage of $40 \times 40°$ for another earthquake of magnitude M7.2 that occurred on 23 October 2011 at 10:41:22 UTC close to the city of Van in eastern Turkey at a shallow depth (10 km) below the shore of Lake Van (38.8°N, 43.5°E). The relationship between the Van earthquake and ATSCTF is

shown in Figure 4. The earthquake occurred during cycle "B" at the time when the $\partial P$ value increased to the maximum amplitude and the $\partial T$ value decreased to the minimum amplitude. The changes in $\partial P$ resulted in an increase in the positive stress of the fault plane, and the changes in $\partial T$ resulted in a decrease in the rupture slide intensity of the fault plane. However, unlike the observations noted for the 2023 earthquake, the changes in $\partial P$ and $\partial T$ were too small to trigger an earthquake (Figure 4). The $\partial N$ value of the ATSCTF was much stronger, and the earthquake occurred at a relatively high value of $\partial N$. This indicates that the ATSCTF was strong enough to enhance the tectonics under conditions of high tectonic stress by disturbing and breaking the balance of the tectonic stress, thereby triggering an earthquake. However, no earthquakes occurred during the other three cycles, not even during cycle "D", for which the $\partial N$ value was higher than it was in cycle "B".

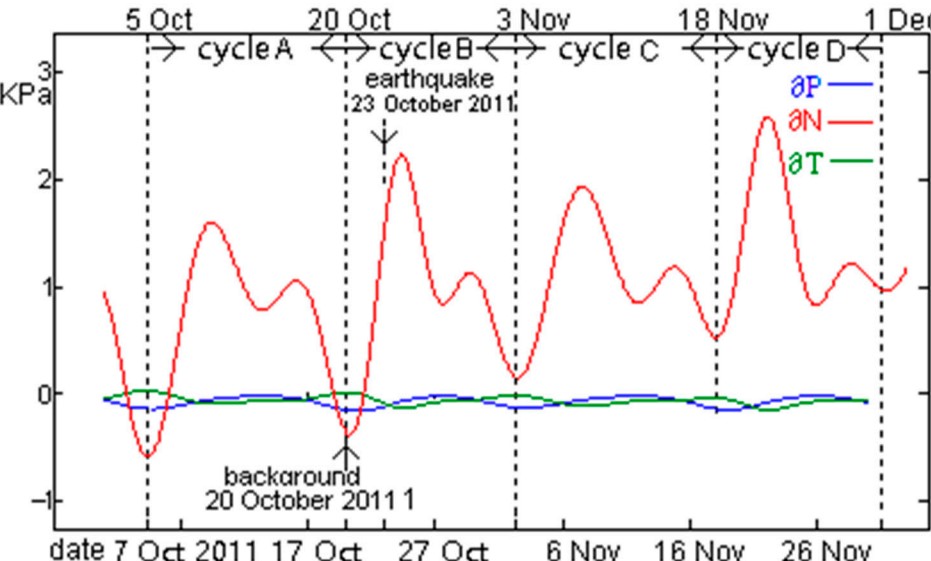

**Figure 4.** The periodic changes in $\partial P$, $\partial T$, and $\partial N$ of the ATSCTF for Turkey's Van earthquake on 23 October 2011.

　　　Figures 2 and 4 show that the 6 February, 2023 M7.8-strong earthquake and 23 October 2011 M7.2-strong earthquake both occurred at the high points (Figures 2 and 4), indicating that the tidal force had a certain triggering effect on the occurrence of earthquakes.

　　　We obtained the spatial and temporal evolution of OLR before, during, and after the earthquake, which correspond to cycles "A", "B", and "C" (Figure 5). The results show that there was no significant enhancement of OLR in and around the epicenter of the Turkish earthquake during the pre-earthquake tidal change cycle "A" and the post-earthquake tidal change cycle "C", and no earthquake occurred. In cycle "B", the OLR changes during the ATSCTF period of the earthquake were clearly shown. On 21 October, the OLR value increased in the north of the epicenter. There was a distinct increase in the OLR value on 22 October, and the OLR area extended mainly to the north of the epicenter. After 22 October, the OLR area shrank around the epicenter. The main shock occurred on 23 October. After the shock was over, the higher OLR value regions were still concentrated at the center of the epicenter. The distribution of anomalous areas was approximately parallel to the geological structure associated with the North Anatolian Fault. The OLR value decreased gradually along the East Anatolian Fault until 25 October (Figure 5). The enhancement demonstrated a similar evolution process (increase→earthquake→shrink around the epicenter) for the 2023 earthquake, and the evolution process was consistent with the rock failure process.

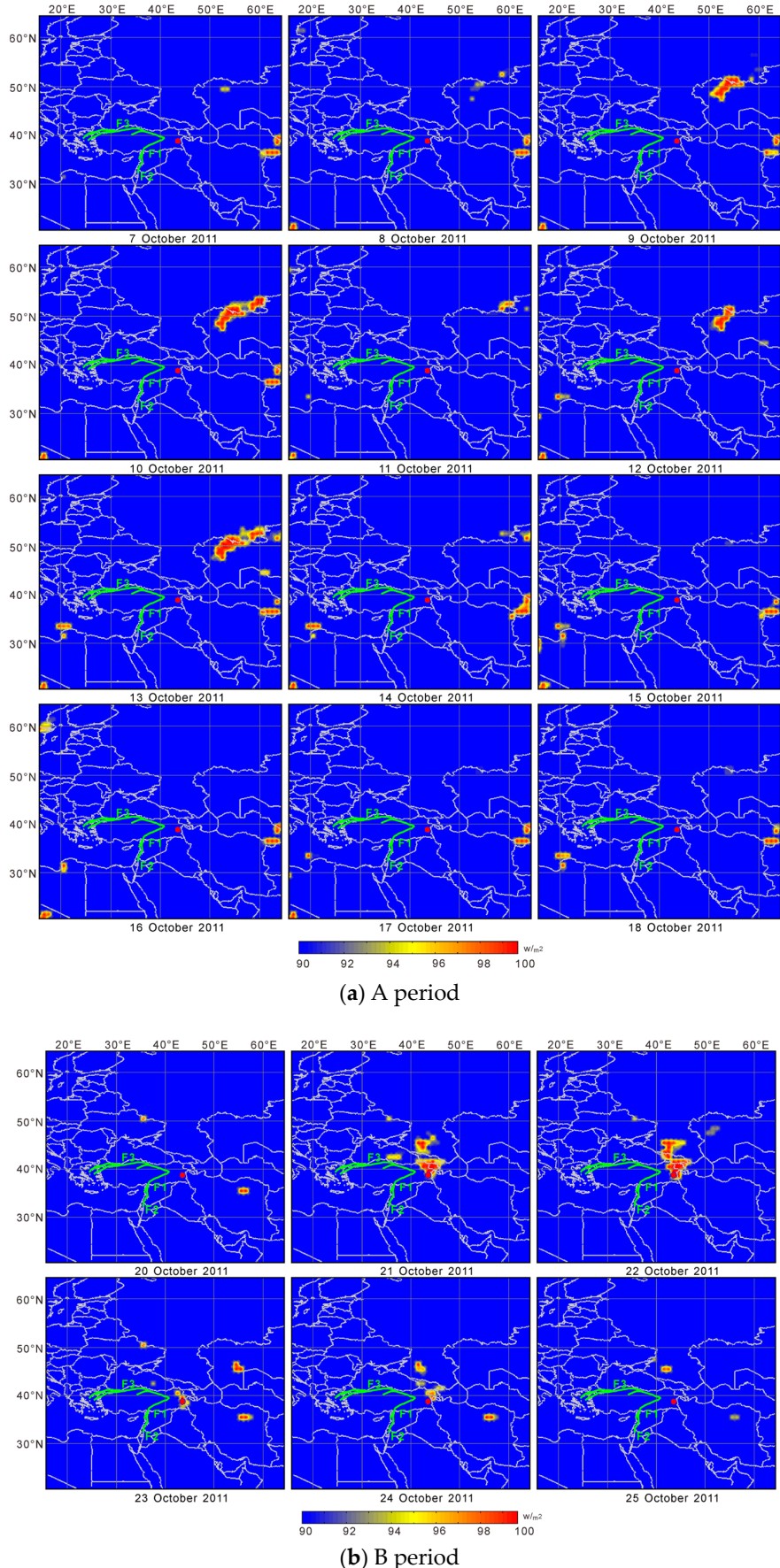

(**a**) A period

(**b**) B period

**Figure 5.** *Cont.*

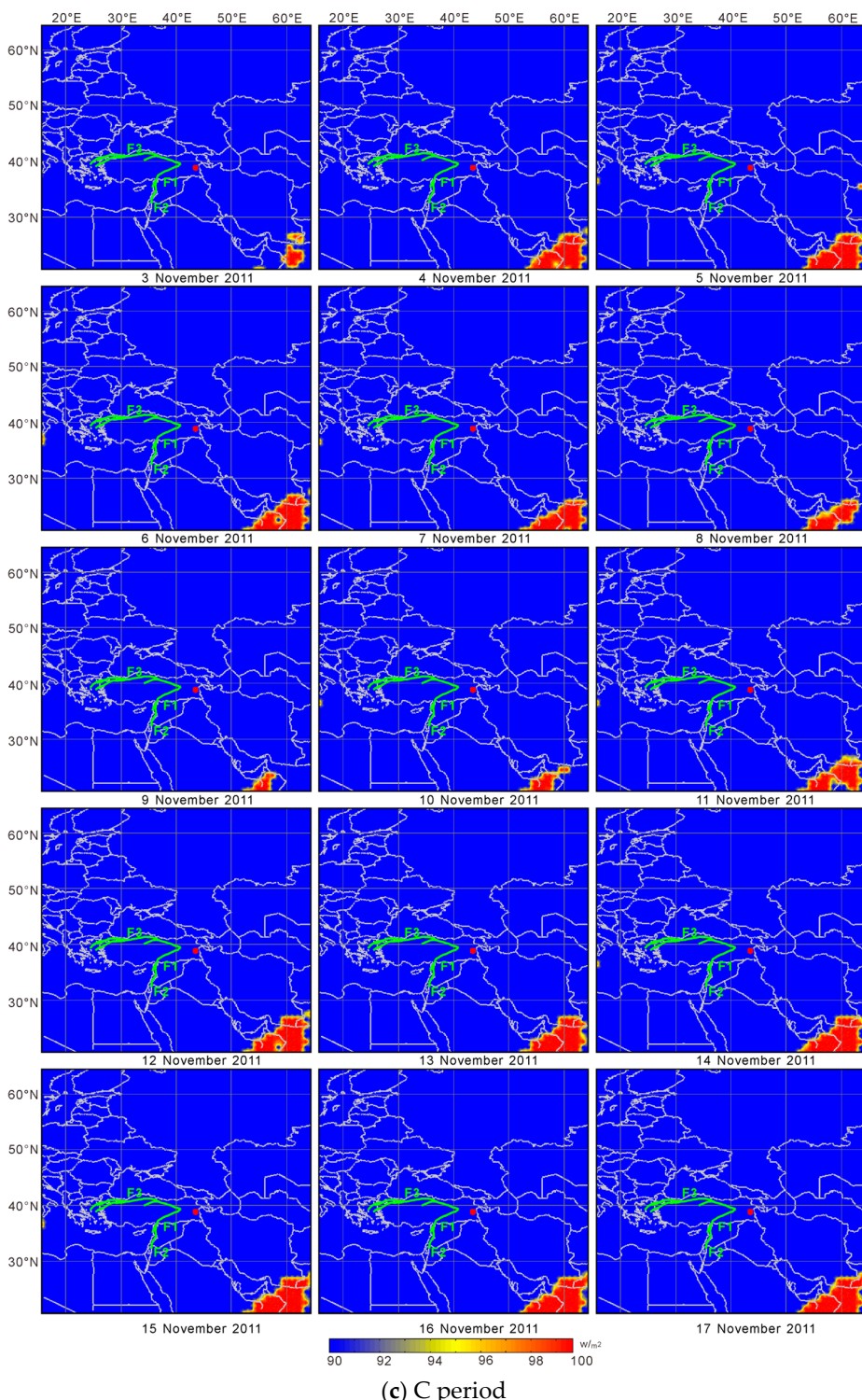

(**c**) C period

**Figure 5.** Incremental field distribution of OLR during the Turkey earthquake on 23 October 2011. The red dots are the epicenters. F1: East Anatolian Fault; F2: North Anatolian Fault; F3: Dead Sea Fault.

## 6. Conclusions

This paper used celestial tide-generating force and OLR data to jointly carry out research on seismic thermal anomaly identification. On the one hand, the change cycle of additive tectonic stress caused by the celestial tide-generating force can provide a clear time indication for the selection of remotely sensed anomaly-monitoring background

data for short-impending earthquakes. At the same time, a mechanical basis for the identification of pre-earthquake thermal anomalies was added to eliminate the uncertainty of the conclusion caused by the randomness of background time selection in statistical processing methods that have existed for many years. On the other hand, the temporal and spatial evolution of OLR anomalies also provide an opportunity to monitor the stress variation of seismogenic structures for tidal force-induced earthquakes. Therefore, the temporal and spatial evolution characteristics of OLR anomalies induced by the celestial tide-generating force and tectonic stress change provide an approach for the research on a short-impending earthquake precursor. It is necessary to further study the physical mechanism of this model of tidal force-induced earthquakes and to explore the regular characteristics of thermal anomalies, so as to improve the capability of monitoring short-impending anomalies of strong earthquakes in specific regions in the future.

**Author Contributions:** Conceptualization, J.L. and J.C.; methodology, J.L., J.C. and Y.Z.; software, J.L.; validation, Y.Z., J.Z., Y.H. and L.W.; formal analysis, J.L.; investigation, J.L.; resources, J.C.; data curation, J.C.; writing—original draft preparation, J.L; writing—review and editing, J.C., Y.Z. and X.S.; visualization, J.C.; supervision, J.C.; project administration, J.C.; funding acquisition, J.C. All authors have read and agreed to the published version of the manuscript.

**Funding:** The work is supported by the National Key Research and Development Project (No. 2021YFB3901203) and the APSCO Earthquake Research Project Phase II.

**Data Availability Statement:** The OLR data of this study can be downloaded from http://www.emc.ncep.noaa.gov (accessed on 5 May 2023).

**Acknowledgments:** Thanks to the NOAA for the OLR data. We also thank all reviewers for their comments and suggestions.

**Conflicts of Interest:** The authors declare no conflict of interest.

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
