# Peer review of "Study of the OLR Anomalies before the 2023 Turkey M7.8 Earthquake"

_remotesensing, doi:10.3390/rs15215078_

Round 1
Reviewer 1 Report
I have read the manuscript with great interest and found that the manuscript is a significant
contribution to the topic only after major revision. There are so many unanswered questions
that need to be addressed before it is ready for publishing.

Minor editing of English language required
Author Response
Dear reviewer,
Thank you for your comments and suggestions. These comments are valuable and very helpful. We have read thorough comments carefully and have made corrections. We uploaded the revised paper and the change-marked-manuscript to the system. Revisions in the text are shown in the change-marked-manuscript. The change-marked-manuscript is after the responses letter. All the reply can be seen in the attachment.

Reviewer 2 Report
This is a fascinating study and very clearly presented. There are only two cases presented, and clearly many more cases will be required to confirm the speculations of the authors, but their conclusion based on these first two cases are warranted based on what they have so far, and I find their arguments toward their conclusion to be very plausible. I think this is a worthy addition to the literature on the subject of earthquake precursor detection and should be published as is. I found only one typo, in line 168, where "tide" is misspelled "tede".
Author Response
Dear reviewer,
Thanks very much for taking your time to review this manuscript and we really appreciate your approval of our article. We have made some modifications based on you and other reviewers in the revised paper. Revisions in the text are shown in the change-marked-manuscript. The change-marked-manuscript is after the responses letter. The spelling of all the words was rechecked and the language was organized by a language editing company. All the changes can be seen in the attachment.

Reviewer 3 Report
This work presents the OLR anomalies prior to the 2023 M7.8 Turkey event. These results are robust. The spatial and temporal relations between the OLR anomalies, earthquakes, and tidal force are quite strong and interesting. However, this work cannot be published before necessary revisions.
Major Comments:
1. The manuscript lacks a lot of details. For example, the method parts are not detailed enough. Moreover, many terminologies and concepts need to be explained. There also seem to be some self-invented terms. These can cause great obstacles and difficulties for readers to understand and read.
2. The anomalies on the EAF are quite inspiring! However, more discussion is necessary. If the thermal anomalies are related to stress, then according to the popular "asperity theory", the thermal anomalies should appear near the asperity, where the stress is the strongest, and the thermal anomalies can also help us to find the asperity. Please refer to the reference [Xu et al., 2023], in this work, the author used the InSar and GPS data to detect the asperity. You may discuss the relation between the spatial location of thermal anomalies and the asperity indicated by other geophysical data. Moreover, as known to all, the locking of the NAF is much stronger than the EAF. This is the reason why I said “The anomalies in the EAF are inspiring”. If it is possible, it would be better to provide more results about the earthquakes occurring on the NAF. If the anomalies related to NAF anomalies are distributed around the NAF, it could be a more valuable result that indicates the potential relation between asperity and thermal anomalies. Please do not forget that, compared to remote sensing, earthquake is more “geophysics” and “seismology”. More discussions are necessary
Minor Comments:
1. Line 19: What is cycle “C”? Please define it and give the date of cycle “C”.
2. Line 21: What are the “non-uniform local heating characteristics”?
3. Line 76: what is the “atmospheric window”? Please define it.
4. Line 82: “National Earthquake Information Center”, which Country?
5. Section 3.2: How do you choose the background? It was not introduced in the method part.
6. Line 199: what is “(3)”?
7. The citation of figures is not in the same style. Sometimes, it is a fig, and sometimes it is a figure. Please correct.
8. The language of this manuscript needs to be improved.
Reference
Xu, L., Y. Aoki, J. Wang, Y. Cui, Q. Chen, Y. Yang, and Z. Yao (2023), The 2023 Mw 7.8 and 7.6 Earthquake Doublet in Southeast Türkiye: Coseismic and Early Postseismic Deformation, Faulting Model, and Potential Seismic Hazard, Seismological Research Letters, doi:10.1785/0220230146.
The organization of statements and the logic of expression is not very “English”. It could be better to invite a native speaker to polish the language.
Author Response
Dear editor,
Thanks very much for taking your time to review this manuscript. We really appreciate all your generous comments and suggestions. All of your questions were answered one by one and we used a professional language company for the language editing. Revisions in the text are shown in the change-marked-manuscript. The change-marked-manuscript is after the responses letter. The responses to the questions and the change-marked-manuscript can be seen inthe attachcment.

Reviewer 4 Report

The overall quality of the English language is fine. But authors are advised to revise the manuscript carefully as there is still some roam for improvements further. Authors may also improve the quality with the help of a native person.
Author Response
Dear reviewer,
Thanks very much for taking your time to review this manuscript. We really appreciate all your generous comments and suggestions. All of your questions were answered one by one and we used a professional language company for the language editing. Revisions in the text are shown in the change-marked-manuscript. The change-marked-manuscript is after the responses letter. The responses can be seen in the attachment.

Round 2
Reviewer 1 Report
The authors have addressed inquiries raised and incorporated points and citations in the paper. Furthermore, they have revised the introduction, discussion, and conclusion parts by adding a few sentences that have enhanced the manuscript. I still have one question related to the response to query 6. The authors mentioned that the tidal stress near the Black Sea fault is different from the tidal stress at the epicenter; therefore, the OLR anomalies obtained in different tidal backgrounds may be different. OLR anomalies can be affected and modulated by various atmospheric parameters. My question is: how does tidal stress affect OLR anomalies? Is there any direct relationship between these two parameters? Is there any specific factors or mechanisms playing role?
Author Response
Dear reviewer,
Thanks very much for taking your time to review the revised manuscript. The responses to the question are shown in the attachment.

Reviewer 4 Report
All responses from the authors are satisfactory. The quality of the manuscript has been improved significantly that can be accepted in the present form.
Author Response
Dear reviewer,
Thank you once again for your valuable suggestions and insightful comments, as well as your endorsement of the revised paper.